# Agnostic Private Density Estimation for GMMs
# via List Global Stability

**Mohammad Afzali**                  AFZALIKM@MCMASTER.CA
*McMaster University*

**Hassan Ashtiani**                   ZOKAEIAM@MCMASTER.CA
*McMaster University*

**Christopher Liaw**                   CVLIAW@GOOGLE.COM
*Google*

**Editors:** Gautam Kamath and Po-Ling Loh

## Abstract

We consider the problem of private density estimation for mixtures of unbounded high-dimensional Gaussians in the agnostic setting. We prove the first upper bound on the sample complexity of this problem. Previously, private learnability of high dimensional GMMs was only known in the realizable setting (Afzali et al., 2024).

To prove our result, we exploit the notion of *list global stability* (Ghazi et al., 2021b,a) that was originally introduced in the context of supervised learning. We define an agnostic variant of this definition, showing that its existence is sufficient for agnostic private density estimation. We then construct an agnostic list globally stable learner for GMMs.

**Keywords:** Differential Privacy, Agnostic Density Estimation, GMMs

## 1. Introduction

Density estimation is a fundamental problem that has been studied for decades in statistics. In this problem, we have access to samples drawn i.i.d. from an unknown distribution $f$ that belongs to a known class of distributions $\mathcal{F}$. The goal is to find a distribution $\hat{f} \in \mathcal{F}$ that is close to $f$ with respect to the total variation distance (i.e., $L_1$ distance). This setting, often referred to as the *realizable* setting, is unrealistic in most cases, since the true distribution might not be exactly a member of the class $\mathcal{F}$. This might happen due to model misspecification or adversarial corruptions. For example, we expect a density estimation method for Gaussians to perform well even if $f$ is only approximately a Gaussian. As opposed to the realizable setting, in the agnostic setting, we do not assume that $f$ belongs to the class $\mathcal{F}$. Instead, the goal is to find a distribution $\hat{f} \in \mathcal{F}$ that is as close to $f$ as possible compared to the best possible distribution in $\mathcal{F}$.

**Definition 1 (Agnostic Density Estimation)** *Let $\mathcal{F}$ be a class of distributions, $C \geq 1$, and $\alpha, \beta \in (0, 1)$. An algorithm $\mathcal{A}$ is said to be a $C$-agnostic $(m, \alpha, \beta)$-learner for $\mathcal{F}$ if:*

> *For every distribution $g$, after receiving an i.i.d. sample set $S$ of size at least $m$ from $g$, the algorithm outputs a distribution $\hat{f} = \mathcal{A}(S)$, such that $\mathrm{d_{TV}}(\hat{f}, g) \leq C \cdot OPT + \alpha$ with probability at least $1 - \beta$, where $\mathrm{d_{TV}}$ is the total variation distance (see Section 2), and $OPT = \inf_{f \in \mathcal{F}} \mathrm{d_{TV}}(g, f)$ measures how far $g$ is from the class $\mathcal{F}$.*

Designing agnostic (and more generally *robust*) density estimators has been the subject of extensive studies in the literature, and several useful tools have been developed for it, such as the Minimum Distance Estimator (Yatracos, 1985; Devroye and Lugosi, 2001) and robust compression schemes (Ashtiani et al., 2020).

We study the problem of agnostic density estimation under the constraint of differential privacy (Dwork et al., 2006b,a), which is the gold standard for protecting individuals' privacy in a dataset. At a high level, differential privacy requires the algorithm's outputs on every two neighbouring datasets to be statistically indistinguishable from each other (see Definition 11).

Given the importance of robustness and privacy, one may ask whether it is possible to design estimators that are both robust and private. In the context of supervised learning, it is known that agnostic supervised learning can often be reduced to learning in the realizable setting (Hopkins et al., 2022a). In fact, Alon et al. (2020) have shown that one can always turn a private classifier that works in the realizable setting into a private classifier that works in the agnostic setting. Therefore, designing sample-efficient agnostic classifiers does not seem to be a particularly challenging task. This picture, however, is completely different in the density estimation setting. For example, it has been shown that there are classes of distributions that are privately learnable in the realizable setting but not in the agnostic setting (Ben-David et al., 2024b). Therefore, there is no general recipe to convert non-robust density estimators to robust ones—either in the private or the non-private setting.

The above observation raises the question of whether there is a framework for designing *private and agnostic* density estimators for the commonly used classes of distributions, such as Gaussians and their mixtures.

For the case of high dimensional Gaussians, private learnability is well understood (Karwa and Vadhan, 2018; Kamath et al., 2019a; Bun et al., 2019a; Biswas et al., 2020; Aden-Ali et al., 2021a; Hopkins et al., 2022b; Kamath et al., 2022b; Ashtiani and Liaw, 2022; Kothari et al., 2022; Alabi et al., 2023; Hopkins et al., 2023). In fact, some of these results establish the private learnability of high dimensional Gaussians with unbounded parameters in the *agnostic setting* (Aden-Ali et al., 2021a; Ashtiani and Liaw, 2022; Kothari et al., 2022; Alabi et al., 2023; Hopkins et al., 2023). See Section 6 for a more thorough discussion of these results and other related work. For the case of GMMs, however, the problem is much more challenging.

## 1.1. Private density estimation for GMMs

Consider the class of mixtures of $k$ Gaussians with arbitrary unbounded parameters (i.e., means, covariances, and mixing weights) in $d$ dimensions. One of the factors that makes private learning of GMMs challenging is the "identifiability" issue: unlike Gaussians, two GMMs with very close densities can have very different parameters. As a result, even non-private *parameter estimation* for GMMs requires an exponential number of samples in terms of the number of components (Moitra and Valiant, 2010). In contrast, non-private *density estimation* for GMMs can be done with a polynomial number of samples in terms of $k$ and $d$ (Devroye and Lugosi, 2001; Ashtiani et al., 2018b,a, 2020). These sample-efficient density estimators are therefore inevitably inaccurate (and practically unstable) in terms of the *parameters* they recover. As a result, some of the standard approaches that are used for private learning of Gaussians do not extend to GMMs. For example, one cannot directly resort to robust-to-private reductions (Hopkins et al., 2023; Asi et al., 2023) since they only work in a finite-dimensional (parameter) space. Similarly, the private to non-private reduction for GMM parameter estimation (Arbas et al., 2023) cannot be directly applied.

Another avenue for design of private density estimators for GMMs is the framework of private hypothesis selection Bun et al. (2019a). In fact, if the Gaussians have bounded parameters and bounded condition numbers then one can build a "finite cover" for GMMs and apply this framework. There is, however, a significant obstacle in extending this approach to GMMs with unbounded parameters. Namely, one would need a "locally small" cover for GMMs, which is hard if not impossible to construct[1].

The private learnability of GMMs in the univariate (and axis-aligned) setting was established by Aden-Ali et al. (2021b) in the realizable setting. The use of stability-based histograms (Bun et al., 2019b) for detecting heavy hitters makes this approach infeasible for handling unbounded high dimensional GMMs.

Recently, the private learnability of high dimensional GMMs has been established in the *realizable setting* (Afzali et al., 2024). One of the main ideas that they exploit is that although parameter estimation for GMMs requires an exponential number of samples, "list decoding" parameters can be done with polynomial number of samples[2]. They then run these list decoders on multiple sub-samples and privately aggregate the results. The aggregation is done via an advanced form of heavy hitter selection in the space of parameters. However, it is not easy to extend their approach to the agnostic setting. In particular, when the samples are heavily corrupted, the lists outputted by list decoders may not contain a approximate heavy hitter. This raises the following question.

> Can mixtures of high dimensional Gaussians with unbounded parameters be privately learned in the agnostic setting using a polynomial number of samples?

In this paper, we resolve the above question by leveraging a form of stability in the design of private algorithms. There are several notions of stability that are related to differential privacy including *global stability* (Thakurta and Smith, 2013; Bun et al., 2020), *(list) replicability* (Impagliazzo et al., 2022; Chase et al., 2023), and *list global stability* (Ghazi et al., 2021a,b). In particular, we find the notion of list global stability suitable for our application[3].

**Definition 2 (List Global Stability (Ghazi et al., 2021a,b))** *For $m, L \in \mathbb{N}$, a list decoding algorithm $\mathcal{A}$ receives a sample $S$ of size $m$ (from an input domain) and outputs a list $H_S$ of size $L$. We say a $\mathcal{A}$ is $(m, \rho, L)$-list-globally-stable if for every distribution $\mathcal{D}$ over input, there exists a hypothesis $h_\mathcal{D}$ such that $\mathbb{P}_{S \sim \mathcal{D}^m} [h_\mathcal{D} \in H_S] \geq \rho$.*

In the context of classification, Ghazi et al. (2021a) describe how to convert a list globally stable learner into a private learner with a number of samples that is logarithmic in $L$. Although this reduction is stated for the realizable setting, it automatically implies an agnostic learner (recall that for private classification, agnostic learnability can be reduced its realizable counterpart (Alon et al., 2020)). However, as discussed before, such a general reduction is impossible for density estimation. Can we still use list globally stability for agnostic and private learning of GMMs?

---

1. Proposition B.6 in Aden-Ali et al. (2021b) demonstrates that a GMMs do not admit a locally small cover the way it is defined in Bun et al. (2019a).

2. Ignoring the components with negligible mixing weights.

3. See Appendix E for a discussion of these notions, and why other notions are not desirable in our setting.

## 1.2. Our contributions

Our contributions are twofold: *(i)* we show a reduction from agnostic private density estimation to list globally stable learning, and *(ii)* we design a list globally stable learner for the class of GMMs, establishing their agnostic private learnability. First, let us explicitly define list globally stable learning in the context of agnostic density estimation.

**Definition 3 (List Global Stability for Agnostic Density Estimation)** *Let $m, L \in \mathbb{N}$, $\alpha \in (0, 1)$, $C > 1$, and $\mathcal{F}$ be class of distributions. We say $\mathcal{A}$ is a $(C, \alpha)$-accurate $(m, \rho, L)$-list-globally-stable learner for $\mathcal{F}$ if for every distribution $g$ (not necessarily in $\mathcal{F}$) there exists distribution $\tilde{g}$ such that*

*(1) $\mathcal{A}$ is a list globally stable algorithm;* $\qquad \mathbb{P}_{S \sim g^m} [\tilde{g} \in \mathcal{A}(S)] \geq \rho$

*(2) $\mathcal{A}$ satisfies agnostic utility guarantee;* $\qquad \mathrm{d_{TV}}(\tilde{g}, g) < \alpha + C. \inf_{f \in \mathcal{F}} \mathrm{d_{TV}}(g, f)$

Our first contribution is to show that given a stable list decoding algorithm for a class of distributions, one can privately learn that class of distributions in the *agnostic* setting (as defined in Def. 1).

**Theorem 4 (Private agnostic learning via list global stability)** *Let $\mathcal{F}$ be a class of distributions. For any $m, L \in \mathbb{N}$, $\alpha, \beta \in (0, 1), C > 1$, if $\mathcal{F}$ is $(C, \frac{\alpha}{3+4C})$-accurate $(m, 0.91, L)$-list-globally-stable learnable, then $\mathcal{F}$ is $(\varepsilon, \delta)$-privately $7C$-agnostic $(n, \alpha, \beta)$-learnable with the following number of samples:*

$$n = \tilde{O}\left(\frac{\log(L/\delta\beta)}{\varepsilon} \cdot (m + \frac{\log(L/\beta)}{\alpha^2})\right)$$

Our reduction is, to some extent, similar to that of Ghazi et al. (2021a) for private classification, but with key differences. At a high level, they run the list globally stable algorithm on sub-samples, identify and remove bad candidates (i.e., those with non-zero empirical error) from each list, and detect repeated candidates in the lists using the *sparse selection* technique (Ghazi et al., 2020). In the agnostic distribution learning setting, however, we cannot filter bad candidates right away. The reason is that whether a candidate is considered "bad" depends on the level of corruption, and the level of corruption is not known and is hard to estimate [4]. Therefore, we instead use a recursive procedure to filter out bad distributions while ensuring (1) privacy, (2) the quality of the remaining candidates, and (3) the existence of at least one repeated good candidate among the remaining candidates. We then use the private selection method of Beimel et al. (2013); Bun et al. (2015) to select a good candidate. An overview of our technique, along with the formal proof and algorithms, is provided in Section 3.

With the reduction of Theorem 4 at hand, the remaining critical question is: can we design an effective list globally stable learner for GMMs? The first observation is that the elements outputted by the list globally stable learner have to be somehow "discretized" otherwise the same element may not repeat exactly (as required by Def. 2 and Def. 3).

Designing a list globally stable learner with good utility is challenging even in the realizable setting. However, a substantially more difficult task is addressing the agnostic setting. Here, the list globally stable algorithm has to be *(1)* "stable" and *(2)* have good utility even when some samples

---

4. For agnostic classification, the error of ERM is a good proxy for the corruption level; but for agnostic density estimation, the corruption level cannot be estimated even without privacy

are corrupted (i.e., when the underlying distribution is not a GMM). With these corruptions, however, the list may not contain any candidate that is "super close" to the underlying GMM. Still, the list should contain (with high probability) a specific "stable" candidate (depending only on the underlying GMM $g$) that is sufficiently close to $g$ (the distance depends on the amount of corruption).

We circumvent the aforementioned challenges and demonstrate that the class of GMMs indeed admits a list globally stable learner (in the agnostic setting). In particular, we prove that (i) Gaussians admit a list globally stable learner (see Lemma 17), and (ii) the class of mixtures of any list globally stable learnable class, is also list globally stable learnable (see Theorem 16). Therefore, we conclude that GMMs are list globally stable learnable and can be integrated into our agnostic private learning framework.

**Theorem 5 (Private agnostic learning GMMs, informal version of 22)** *Let $\alpha, \delta \in (0, 1)$ and $\varepsilon \geq 0$. The class of mixtures of $k$ unbounded $d$ dimensional Gaussians is $(\varepsilon, \delta)$-privately 21-agnostic $(n, \alpha, 0.99)$-learnable with*

$$n = \tilde{O}\left(\frac{k^2 d^4 \log(1/\delta)}{\alpha^2 \varepsilon}\right).$$

It is worth mentioning that our agnostic result also improves the realizable result of Afzali et al. (2024) in terms of sample complexity by a factor of $1/\alpha^2$, and achieves optimal dependence on accuracy parameter $\alpha$.

### 1.3. Paper organization

We define some notations in Section 2 before stating the formal proofs. Section 3 provides the high-level proof idea. The formal proof of the main reduction (Theorem 4) is given in Appdendix A. In Section 4, we develop a useful tool for list globally stable learning mixture distributions. In Section 5, we prove the stable list global stability of Gaussians and their mixtures, and conclude with the agnostic private learnability of GMMs. Finally, we review some related work in Section 6.

## 2. Preliminaries

For a set $\mathcal{F}$, define $\mathcal{F}^k = \mathcal{F} \times \cdots \times \mathcal{F}$ ($k$ times), and $\mathcal{F}^* = \bigcup_{k=1}^{\infty} \mathcal{F}^k$. For two absolutely continuous densities $f_1(x), f_2(x)$ on $\mathbb{R}^d$, the total variation (TV) distance is defined as $d_{\mathrm{TV}}(f_1, f_2) = \frac{1}{2}\int_{\mathbb{R}^d} |f_1(x) - f_2(x)|\, dx$. In this paper, if $\mathrm{dist}: \mathcal{F} \times \mathcal{F} \to \mathbb{R}_{\geq 0}$ is a metric, $f \in \mathcal{F}$, and $\mathcal{F}' \subseteq \mathcal{F}$ then we define $\mathrm{dist}(f, \mathcal{F}') = \inf_{f' \in \mathcal{F}'} \mathrm{dist}(f, f')$.

**Definition 6 ($\alpha$-cover)** *A set $C_\alpha \subseteq \mathcal{F}$ is said to be an $\alpha$-cover for a metric space $(\mathcal{F}, \mathrm{dist})$, if for every $f \in \mathcal{F}$, we have $\mathrm{dist}(f, C_\alpha) \leq \alpha$.*

**Definition 7 ($k$-mixtures)** *Let $\mathcal{F}$ be an arbitrary class of distributions. We denote the class of $k$-mixtures of $\mathcal{F}$ by $k\text{-mix}(\mathcal{F}) = \Delta_k \times \mathcal{F}^k$ where $\Delta_k = \{w \in \mathbb{R}^k : w_i \geq 0, \sum_{i=1}^{k} w_i = 1\}$ is the $(k-1)$-dimensional probability simplex.*

In this paper, we simply write $f \in k\text{-mix}(\mathcal{F})$ to denote a $k$-mixture from the class $\mathcal{F}$ where the representation of the weights is implicit. Further, if $f \in k_1\text{-mix}(\mathcal{F})$ and $g \in k_2\text{-mix}(\mathcal{F})$ then we write $d_{\mathrm{TV}}(f, g)$ to denote the TV distance from the underlying distribution.

**Definition 8 (Unbounded Gaussians)** *Let $\mathcal{G}_d = \{\mathcal{N}(\mu, \Sigma) : \mu \in \mathbb{R}^d, \Sigma \in S^d\}$ be the class of d-dimensional Gaussians, where $S^d$ is the set of positive-definite cone in $\mathbb{R}^{d \times d}$.*

The following result on learning a finite class of distributions is based on the Minimum Distance Estimator (Yatracos, 1985); see the excellent book by Devroye and Lugosi (2001) for details.

**Theorem 9 (Learning finite classes, Theorem 6.3 of Devroye and Lugosi (2001))** *Let $\mathcal{F}$ be a finite class of distributions, and $\alpha, \beta \in (0, 1)$. Then, $\mathcal{F}$ is 3-agnostic $(n, \alpha, \beta)$-learnable with $n = O(\frac{\log |\mathcal{F}| + \log(1/\beta)}{\alpha^2})$.*

## 2.1. Differential privacy

Two datasets $D, D' \in \mathcal{X}^n$ are called neighbouring datasets if they differ by one element. We use the notation $D \sim D'$ to denote that they are neighboring datasets. Informally, a differentially private algorithm is required to have close output distributions on neighbouring datasets.

**Definition 10 ($(\varepsilon, \delta)$-Indistinguishable)** *Two distribution $f_1, f_2$ with support $\mathcal{X}$ are said to be $(\varepsilon, \delta)$-indistinguishable if for all measurable subsets $E \in \mathcal{X}$, $\mathbb{P}_{X \sim f_1}[X \in E] \leq e^{\varepsilon} \mathbb{P}_{X \sim f_2}[X \in E] + \delta$ and $\mathbb{P}_{X \sim f_2}[X \in E] \leq e^{\varepsilon} \mathbb{P}_{X \sim f_1}[X \in E] + \delta$.*

**Definition 11 ($(\varepsilon, \delta)$-Differential Privacy (Dwork et al., 2006b,a))** *A randomized algorithm $\mathcal{M} : \mathcal{X}^n \to \mathcal{Y}$ is said to be $(\varepsilon, \delta)$-differentially private if for every two neighbouring datasets $D, D' \in \mathcal{X}^n$, the output distributions $\mathcal{M}(D), \mathcal{M}(D')$ are $(\varepsilon, \delta)$-indistinguishable.*

We utilize the property of differentially private algorithms that guarantees the preservation of differential privacy when these algorithms are composed adaptively. By adaptive composition, we refer to executing a series of algorithms $\mathcal{M}_1(D), \mathcal{M}_2(D), \cdots, \mathcal{M}_k(D)$, where the selection of the algorithm $\mathcal{M}_i$ can depend on the outputs of the preceding algorithms $\mathcal{M}_1(D), \mathcal{M}_2(D), \cdots, \mathcal{M}_{i-1}(D)$.

**Lemma 12 (Composition of DP (Dwork et al., 2006b,a))** *If algorithms $\mathcal{M}_1, \mathcal{M}_2, \cdots, \mathcal{M}_k$ are $(\varepsilon_1, \delta_1), (\varepsilon_2, \delta_2), \cdots, (\varepsilon_k, \delta_k)$-differentialy private, and $\mathcal{M}$ is an adaptive composition of $\mathcal{M}_1, \cdots, \mathcal{M}_k$, then $\mathcal{M}$ is $(\sum_{i \in [k]} \varepsilon_i, \sum_{i \in [k]} \delta_i)$-DP.*

Consider the problem of private selection, where we are given a set of candidates and a score function to measure how "good" each candidate is with respect to a given data set. A well known method for privately selecting a "good" candidate is the Exponential Mechanism (McSherry and Talwar, 2007).

Nonetheless, the Exponential Mechanism may not have a good utility when the number of candidates is very large or infinite. Another useful tool for this task, is therefore the Choosing Mechanism of Beimel et al. (2013); Bun et al. (2015), which is compatible with infinite candidate sets. The Choosing Mechanism ensures returning a "good" candidate, as long as the score function has a bounded growth.

**Theorem 13 (Choosing Mechanism, Lemma 3.8 of Bun et al. (2015))** *Let $(\mathcal{F}, \kappa)$ be a metric space and $\mathcal{X}$ be an arbitrary set. Let* score$: \mathcal{F} \times \mathcal{X}^T \to \mathbb{R}_{\geq 0}$ *be a function such that:*

- score$(f, \emptyset) = 0$ *for all $f \in \mathcal{F}$.*

---

**Algorithm 1:** Choosing Mechanism

---

**Input:** $D \in \mathcal{X}^T$, candidate set $\mathcal{F}$, quality function score: $\mathcal{F} \times \mathcal{X}^T \to \mathbb{R}_{\geq 0}$, parameters $\beta, \varepsilon, \delta, k$.

1 Set $\text{MAX} = \max_{f \in \mathcal{F}} \text{score}(f, D)$;

2 If $\widetilde{\text{MAX}} \leq \frac{8}{\varepsilon} \log(\frac{4k}{\beta \varepsilon \delta})$, reject and return $\perp$ ;

3 Set $G = \{f \in \mathcal{F} : \text{score}(f, D) \geq 1\}$ ;

4 **return** $\hat{f} \in G$ with probability $\propto \exp(\varepsilon \cdot \text{score}(\hat{f}, D)/4)$.

---

- *If $D' = D \cup \{x\}$, then $\text{score}(f, D) + 1 \geq \text{score}(f, D') \geq \text{score}(f, D)$ for all $f \in \mathcal{F}$.*

- *There are at most $k$ values of $f \in \mathcal{F}$ such that $\text{score}(f, D) + 1 = \text{score}(f, D')$.*

*Then algorithm 1 is $(\varepsilon, \delta)$-DP with the following property. For every $D \in \mathcal{X}^T$, with probability at least $1 - \beta$, it outputs $\hat{f} \in \mathcal{F}$ satisfying:*

$$\text{score}(\hat{f}, D) \geq \max_{f \in \mathcal{F}} \text{score}(f, D) - \frac{16}{\varepsilon} \log(\frac{4kT}{\beta \varepsilon \delta}).$$

We will also use the Truncated Laplace distribution, which will be useful for privately answering threshold queries in our algorithm.

**Definition 14 (Truncated Laplace distribution)** *Let $\delta \in (0, 1)$, and $\varepsilon, \Delta > 0$. Truncated Laplace distribution is denoted by $TLap(\Delta, \varepsilon, \delta)$ with the following density function:*

$$f_{TLap(\Delta, \varepsilon, \delta)}(x) = \begin{cases} \frac{\varepsilon}{2\Delta(1 - e^{-\varepsilon R/\Delta})} e^{-\varepsilon|x|/\Delta} & x \in [-R, R] \\ 0 & x \notin [-R, R]. \end{cases}$$

*where $R = \frac{\Delta}{\varepsilon} \log(1 + \frac{e^{\varepsilon} - 1}{2\delta})$.*

**Lemma 15 (Theorem 1 of Geng et al.)** *Let $\delta \in (0, 1), \varepsilon > 0$, and $q : \mathcal{X}^n \to \mathbb{R}^d$ be a function, define the sensitivity of $q$ to be $\Delta := \max_{D \sim D' \in \mathcal{X}^n} ||q(D) - q(D')||_1$. Then $q(x) + Y$ is $(\varepsilon, \delta)$-DP, where $Y \sim TLap(\Delta, \varepsilon, \delta)$.*

## 3. From list global stability to agnostic private density estimation

In this section we provide a high level proof overview of Theorem 4. We show that if a class of distributions is list globally stable learnable, then it is privately learnable in the agnostic setting. Let us restate the theorem.

**Theorem 4 (Private agnostic learning via list global stability)** *Let $\mathcal{F}$ be a class of distributions. For any $m, L \in \mathbb{N}, \alpha, \beta \in (0, 1), C > 1$, if $\mathcal{F}$ is $(C, \frac{\alpha}{3 + 4C})$-accurate $(m, 0.91, L)$-list-globally-stable learnable, then $\mathcal{F}$ is $(\varepsilon, \delta)$-privately $7C$-agnostic $(n, \alpha, \beta)$-learnable with the following number of samples:*

$$n = \tilde{O}\left(\frac{\log(L/\delta\beta)}{\varepsilon} \cdot (m + \frac{\log(L/\beta)}{\alpha^2})\right)$$

First, we give a high-level overview of Algorithm 2. The procedure consists of three steps:

**Step I.** We split the dataset (drawn i.i.d. from an unknown distribution $g$) into $T$ disjoint subsets. We run the list globally stable learner for $\mathcal{F}$ on each subset and get $T$ lists of distributions. Given the definition of the list globally stable learner (see Definition 3), we can conclude that there exists a distribution $\tilde{g}$ which is in a large fraction of lists with high probability. In other words, $\tilde{g}$ is a "stable" output.

**Step II.** Next, we filter the lists and ensure that all distributions are close to the true distribution $g$ w.r.t. $d_{\mathrm{TV}}$. This can be done using the MDE algorithm (see Theorem 9). Within each list, we find a distribution $\hat{g}$ that is close to the true distribution $g$ w.r.t. $d_{\mathrm{TV}}$, and remove any other distribution that is far from $\hat{g}$. The filtering should be done in a way that does not remove all "stable" distributions from the lists (one of which is $\tilde{g}$). Thus, the filtering radius needs to be chosen carefully since we are not aware of the distance $d_{\mathrm{TV}}(g, \mathcal{F})$. We will describe how to privately choose a suitable filtering radius using binary search combined with the Propose-Test-Release framework (see Algorithm 3).

**Step III.** Finally, we consider the candidate set $\mathcal{M}$ to be the union of all filtered lists (which is a subset of $\mathcal{F}$) and assign a score to each candidate to measure how "stable" that candidate is w.r.t. the lists. We know there exists at least one stable candidate which is not filtered from the lists since $\mathcal{A}$ is an algorithm that "preserves agnostic utility guarantee" (see Definition 3). Also, we know that the size of each list is bounded, which allows us to conclude there are not too many "stable" candidates. In other words, we design the score function to take advantage of this fact and have a bounded growth (in the sense of Theorem 13). This allows us to privately select a "stable" candidate using the Choosing Mechanism (see Theorem 13). The second (filtering) step ensures that the selected candidate is close to $g$ w.r.t. $d_{\mathrm{TV}}$.

The formal proof is given in Appendix A.

## 4. List globally stable learning of mixture distributions

In this section, we develop a tool for list globally stable learning mixture distributions. At a high level, we show that if a class of distributions is list globally stable learnable, then the class of its mixtures is also list globally stable learnable. This task is challenging since some components of the mixture might be heavily corrupted, while others may have negligible weights and be difficult to recover.

**Theorem 16** *Let $\mathcal{F}$ be a class of distributions, $\alpha, \beta \in (0, 1)$, $C > 1$, and $L, m \in \mathbb{N}$. If $\mathcal{F}$ is $(C, \alpha)$-accurate $(m, 1 - \beta, L)$-list-globally-stable learnable, then $k$-mix$(\mathcal{F})$ is $(C, 5\alpha)$-accurate $(m_1, 1 - 2k\beta, L_1)$-list-globally-stable learnable, where $L_1 = (\frac{Lk}{\alpha})^k \left( \frac{10ek \log(1/\beta)}{\alpha} \right)^{mk}$, and $m_1 = \frac{2mk + 8k \log(1/\beta)}{\alpha}$.*

Here, we give a high level idea of the proof. First, we show that it is possible to represent the true distribution $g$ as a mixture distribution such that the total mass of its "far" components from $\mathcal{F}$ is small. Now, given a large enough sample from $g$ we can hope that we receive some samples from all non-negligible components of $g$. Assuming we have access to a list globally stable learner algorithm $\mathcal{A}_1$ for $\mathcal{F}$, we can apply it on every subset of samples, and approximately recover each "non-far" component in $g$. We can then compose all the outputs of the algorithm $\mathcal{A}_1$ to create a list globally stable learner $\mathcal{A}_2$ for $k$-mix$(\mathcal{F})$. We show there exist a distribution $\tilde{g}$ in the output of

algorithm $\mathcal{A}_2$ such that it satisfies two properties of Definition 3. In other words, we show that *(1)* $\mathcal{A}_2$ is a list globally stable algorithm, and *(2)* $\mathcal{A}_2$ satisfies agnostic utility guarantee.

The formal proof is given in Appendix B.

## 5. Agnostic private learning of GMMs

As the main application of our framework, we prove the first sample complexity upper bound for privately learning GMMs in the agnostic setting. It is notable that our sample complexity also improves the *realizable* result of Afzali et al. (2024) in terms of accuracy parameter by a factor of $\frac{1}{\alpha^2}$.

Given our general reduction in Section 3, it is sufficient to show that GMMs are list globally stable learnable. Indeed, we show that Gaussians are list globally stable learnable and use our the tool from Section 4 to conclude Gaussian mixtures are also list globally stable learnable.

### 5.1. List globally stable learning of Gaussians and their mixtures

We create a list globally stable learner for the class of Gaussians using the robust sample compression schemes of Ashtiani et al. (2020). At a high level, a class of distributions admits a compression scheme if upon receiving some samples from an unknown distribution, there exists a small subset of those samples that can be used to recover the original distribution up to a reasonable error w.r.t. $\mathrm{d_{TV}}$. At a high level, given a set of samples from an unknown distribution $g$, we run the robust compression algorithm (from Lemma 19) on every subset of samples. Let $g^* = \arg\min_{g' \in \mathcal{G}_d} \mathrm{d_{TV}}(g, g')$. If $g$ is not too far from the class $\mathcal{G}_d$ (e.g. $\mathrm{d_{TV}}(g, g^*) \leq \frac{1}{3}$), then $g^*$ can be approximately recovered and is considered a stable candidate. Therefore, it is sufficient to output all discretized neighbors of the recovered elements to include stable elements in the outputted list.

**Lemma 17 (List globally stable learning of Gaussians)** *Let $\alpha, \beta \in (0,1)$, then $\mathcal{G}_d$ is $(3, \alpha)$-accurate $(m, 1 - \beta, L)$-list-globally-stable learnable, where $L = (d \log(1/\beta))^{O(d^2 \log(1/\alpha))}$, and $m = O(d \log(1/\beta))$.*

The formal definition of the robust compression schemes is given below.

**Definition 18 (Definition 4.2 of Ashtiani et al. (2020), Robust compression)** *Let $\tau, t, m : (0,1) \to \mathbb{Z}_{\geq 0}$ be functions, $\mathcal{F}$ be a class of distributions, and $r \geq 0$. We say $\mathcal{F}$ is $r$-robust $(\tau, t, m)$-compressible, if there exists an algorithm $\mathcal{A}$ such that for any $\alpha, \beta \in (0,1)$, any $f \in \mathcal{F}$, and any distribution $g$ satisfying $\mathrm{d_{TV}}(g, f) \leq r$ the following holds:*

*Let $S$ be an i.i.d. sample set from $g$ of size $m(\alpha) \log(1/\beta)$. Then there exists a sequence $L$ of at most $\tau(\alpha)$ samples from $S$, and a sequence $B$ of at most $t(\alpha)$ bits, such that the algorithm $\mathcal{A}(L, B)$ outputs a distribution satisfying $\mathrm{d_{TV}}(\mathcal{A}(L, B), f) \leq \alpha$ with probability at least $1 - \beta$.*

The next lemma asserts that the class of Gaussians is robust compressible. We will later use this compressing algorithm in order to construct a list globally stable learner for Gaussinas.

**Lemma 19 (Lemma 5.3 of Ashtiani et al. (2020), Robust Compressing Gaussians)** *Let $\alpha \in (0,1)$, then the class $\mathcal{G}_d$ is $\frac{1}{3}$-robust $\left(O(d), O(d^2 \log(d/\alpha)), O(d)\right)$-compressible.*

In order to create a list globally stable learner for a class of distributions, we need the elements outputted by the algorithm to be somehow "discretized", the following lemma will later be useful to do so.

**Lemma 20 (Lemma 8.2 of Afzali et al. (2024))** *For any $0 < \alpha \leq \frac{1}{600}$, there exists an $\alpha$-$d_{\mathrm{TV}}$-cover $\mathcal{C}$ for the class $\mathcal{G}_d$ satisfying:*

$$\sup_{g \in \mathcal{G}_d} |\{g' \in \mathcal{C} : d_{\mathrm{TV}}(g', g) \leq 2\alpha\}| \leq 2^{O(d^2)}.$$

Putting together, we construct a list globally stable learner for the class of Gaussians.

**Proof of Lemma 17:**

**Proof** Using Lemma 19, we know there is a $\frac{1}{3}$-robust $(\tau, t, m')$-compressing algorithm $\mathcal{A}_1$ for $\mathcal{G}_d$, where $\tau = O(d)$, $t = O(d^2 \log(d/\alpha))$, and $m' = O(d)$. Also, let $\mathcal{C}$ be the $\alpha$-$d_{\mathrm{TV}}$-cover from Lemma 20, $g$ be the true distribution, and $g_0 \in \mathcal{C}$ be a dummy distribution. Let $S$ be an i.i.d. sample set of size $m = m' \log(1/\beta)$ from $g$. Consider the set $\mathcal{H}_1 = \{\mathcal{A}_1(S', B) : S' \subseteq S, |S'| \leq \tau, B \in \{0, 1\}^t\}$. Next, construct $\mathcal{H}_2 = \{g' \in \mathcal{C} : \exists h \in \mathcal{H}_1 \text{ s.t. } d_{\mathrm{TV}}(g', h) \leq 2\alpha\} \cup \{g_0\}$. We claim that the algorithm $\mathcal{A}_2$ that takes $S$ as input, and outputs $\mathcal{H}_2$, is a $(3, \alpha)$-accurate $(m, 1 - \beta, L)$-list-globally-stable learner for $\mathcal{G}_d$. To prove this we need to show that for every $g$ there exists a distribution $\tilde{g}$ satisfying the properties of Definition 3. Consider the following two cases:

**Case 1.** If $d_{\mathrm{TV}}(g, \mathcal{G}_d) > \frac{1}{3}$. Consider $\tilde{g} = g_0$. Then, it holds that:

*(1)* $\tilde{g} \in \mathcal{H}_2$, with probability 1. Since by construction, $g_0$ is always in $\mathcal{H}_2$.
*(2)* $d_{\mathrm{TV}}(\tilde{g}, g) \leq 1 < 3 \cdot d_{\mathrm{TV}}(g, \mathcal{G}_d)$

**Case 2.** If $d_{\mathrm{TV}}(g, \mathcal{G}_d) \leq \frac{1}{3}$, let $g^* = \arg\min_{g' \in \mathcal{G}_d} d_{\mathrm{TV}}(g, g')$. Using the definition of robust compression, we know that since $d_{\mathrm{TV}}(g, g^*) \leq \frac{1}{3}$, there exists a distribution $\hat{g} \in \mathcal{H}_1$, satisfying $d_{\mathrm{TV}}(\hat{g}, g^*) \leq \alpha$. Now, let $\tilde{g} = \arg\min_{g' \in \mathcal{C}} d_{\mathrm{TV}}(g^*, g')$. Then it holds that:

*(1)* $\tilde{g} \in \mathcal{H}_2$, with probability at least $1 - \beta$.
Since, $d_{\mathrm{TV}}(\tilde{g}, \hat{g}) \leq d_{\mathrm{TV}}(\tilde{g}, g^*) + d_{\mathrm{TV}}(g^*, \hat{g}) = 2\alpha$ (Recall that $\mathcal{C}$ is an $\alpha$-cover).
*(2)* $d_{\mathrm{TV}}(\tilde{g}, g) \leq d_{\mathrm{TV}}(\tilde{g}, g^*) + d_{\mathrm{TV}}(g^*, g) \leq \alpha + d_{\mathrm{TV}}(g, \mathcal{G}_d)$.

Furthermore, we have $|\mathcal{H}_2| \leq |\mathcal{H}_1| \cdot (\sup_{h \in \mathcal{H}_1} |\{g' \in \mathcal{C} : d_{\mathrm{TV}}(g', h) \leq 2\alpha\}|) \leq O(m^{\tau+t}) \cdot 2^{O(d^2)} = L$, where the last inequality follows from Lemma 20, and concludes the desired result. ∎

The formal proof is given in Section 5.1. An immediate corollary of Lemma 17 and Theorem 16 is that the class of GMMs is list globally stable learnable.

**Corollary 21** *Let $\alpha, \beta \in (0, 1)$, then $k$-mix$(\mathcal{G}_d)$ is $(3, 5\alpha)$-accurate $(m_1, 1 - 2k\beta, L_1)$-list-globally-stable learnable, where $L_1 = (\frac{Lk}{\alpha})^k \left(\frac{10ek \log(1/\beta)}{\alpha}\right)^{mk}$, $m_1 = \frac{2mk + 8k \log(1/\beta)}{\alpha}$, $L = (d \log(1/\beta))^{O(d^2 \log(d/\alpha))}$, and $m = O(d \log(1/\beta))$.*

## 5.2. Agnostic private learning of GMMs

In the following theorem provide the first sample complexity upper bound for agnostic private learning GMMs. The proof (stated in C) is the direct result of Theorem 22 and Corollary 21.

**Theorem 22 (Private agnostic learning GMMs)**  *Let $\alpha, \beta, \delta \in (0,1)$ and $\varepsilon \geq 0$. The class $k$-mix$(\mathcal{G}_d)$ is $(\varepsilon, \delta)$-privately 21-agnostic $(n, \alpha, \beta)$-learnable with*

$$n = \tilde{O}\left(\frac{k^2 d^4 + k d^2 \log(1/\delta\beta) + \log^2(1/\beta)}{\alpha^2 \varepsilon}\right).$$

It is worth mentioning that our algorithm is not computationally efficient. Designing an efficient algorithm for learning GMMs, even in the non-private setting, remains an important open question (Diakonikolas et al., 2017).

## 6. More on related work

In this section, we provide some related work on stability, private distribution learning, and privately learning Gaussian distributions and their mixtures.

The notion of global stability was introduced by Bun et al. (2020) to show the equivalence between online learnability and private learnability of a given concept class. This notion was later refined by Ghazi et al. (2021a,b). Gloabl stability was further studied in connection with algorithmic replicability (Chase et al., 2023; Kalavasis et al., 2023).

Unlike classification, in the distribution learning setting, PAC learnability of general classes of distributions (even in the non-private setting) remains an important open question (Diakonikolas, 2016). Recently, Lechner and Ben-David (2023) showed that there is no single notion of dimension that characterizes the learnability of a given class of distributions.

Another difference between learning distributions and learning concept classes is discussed in Ben-David et al. (2024b). They show that, unlike classification where realizable and agnostic learning is characterized by the VC dimension of the concept class, there is a class of distributions that is learnable in the realizable setting but not in the agnostic setting.

A recent relevant work of Bun et al. (2023) also uses stability-based techniques for private and replicable agnostic-to-realizable reductions for classification. They use all possible labelings of samples to reduce the agnostic replicable to the realizable setting. However, this is again not applicable in the density estimation setting.

Given the connections between robustness and private statistical estimation (Dwork and Lei, 2009; Georgiev and Hopkins, 2022; Liu et al., 2022; Hopkins et al., 2023; Asi et al., 2023; Liu et al., 2021), it is a natural question to ask if every agnostic learnable class of distributions can be learned privately (Afzali et al., 2024). This conjecture is more likely to hold in the agnostic setting, since there is a partial resolution in Bun et al. (2024) stating that there is a class of distributions that can be learned in the realizable setting with a constant accuracy, but not privately learned with the same level of accuracy.

There is a long line of work trying to demonstrate the private learnability of known classes of distributions, such as Gaussians and their mixtures. Karwa and Vadhan (2018) presented the first result on the private learnability of unbounded univariate Gaussians. Later, this result was extended to high-dimensional Gaussians with bounded parameters (Kamath et al., 2019a; Biswas et al., 2020; Hopkins et al., 2022b) and unbounded parameters (Aden-Ali et al., 2021a).

In the unbounded setting, although the result of Aden-Ali et al. (2021a) was nearly tight, matching the lower bound of Kamath et al. (2022a), it was computationally inefficient. This was later improved in Kamath et al. (2022b); Kothari et al. (2022); Ashtiani and Liaw (2022), with the method of Ashtiani and Liaw (2022) achieving near-optimal sample complexity. The results of Kothari et al.

(2022) and Ashtiani and Liaw (2022) also apply in the robust setting with sub-optimal sample complexity. In the robust setting, the later work of Alabi et al. (2023) improved the sample complexity in terms of dependence on the dimension. Recently, Hopkins et al. (2023) achieved a robust and efficient learner with near-optimal sample complexity for unbounded Gaussians. There are also lower bounds on the sample complexity of private statistical estimations related to Gaussians (Portella and Harvey, 2024; Narayanan, 2023; Kamath et al., 2022a; Bun et al., 2014).

There has been an extensive line of research on parameter learning and density estimation of Gaussian Mixture Models (GMMs). The goal of parameter learning is to recover the underlying unknown parameters of the GMM, whereas the goal of density estimation is to find a distribution that closely approximates the underlying distribution with respect to $\mathrm{d_{TV}}$. For the parameter learning task (even in the non-private setting), the exponential dependence of the sample complexity on the number of components is inevitable (Moitra and Valiant, 2010).

There are several works in the private parameter estimation setting for GMMs (Nissim et al., 2007; Vempala and Wang, 2004; Chen et al., 2023; Kamath et al., 2019b; Achlioptas and McSherry, 2005; Cohen et al., 2021; Bie et al., 2022; Arbas et al., 2023).

Unlike parameter estimation, the sample complexity for density estimation can be polynomial in the number of components. In the non-private setting, several results have addressed the sample complexity of learning GMMs (Devroye and Lugosi, 2001; Ashtiani et al., 2018b), culminating in the work by Ashtiani et al. (2018a, 2020) that provides the near-optimal bound of $\tilde{\Theta}(kd^2/\alpha^2)$.

In the private setting, one approach would be to create a locally small cover for the class of GMMs and apply the private hypothesis selection method of Bun et al. (2019a). However, this turns out to be impossible, as Aden-Ali et al. (2021b) showed that the class of GMMs does not admit a locally small cover. They introduced the first polynomial sample complexity upper bound for learning unbounded axis-aligned GMMs under the constraint of approximate differential privacy (DP). They extended the concept of stable histograms from Karwa and Vadhan (2018) to learn univariate GMMs. However, this approach cannot be generalized to general GMMs, as it remains unclear how to learn even a single high-dimensional Gaussian using a stability-based histogram.

Recently, Ben-David et al. (2024a) proposed a pure DP method for learning general GMMs, assuming they have access to additional public samples.

Finally, Afzali et al. (2024) proposed the first polynomial sample complexity upper bound of $\tilde{O}(\frac{k^2 d^4 \log(1/\delta)}{\alpha^4 \epsilon})$ for privately learning general GMMs in the realizable setting. They show that if one has access to a locally small cover and a list decoding algorithm for a class of distributions (e.g., Gaussians), then mixtures of that class (e.g., GMMs) can be learned privately in the realizable setting. At a high level, a locally small cover is an accurate cover that has a small doubling dimension (is not too dense). A list decoding algorithm is an algorithm that receives some sample from an unknown distribution $g$ and outputs a short list of distributions $\mathcal{L}$, one of which is very close to $g$. This latter condition is hard to satisfy in the agnostic setting since one cannot hope to recover a heavily corrupted distribution up to a very small error. Moreover, constructing a locally small cover is a delicate matter for the class of high-dimensional distributions (e.g., Gaussians). In contrast, we consider the distinct notion of list global stability and show that it is enough for privately learning a class (even in the agnostic setting). Using this new notion and reduction has two benefits: (1) the underlying class does not need to admit a locally small cover, and (2) there is no need to have an accurate list decoding algorithm (which is not possible when the unknown distribution is heavily corrupted). Moreover, we come up with a list globally stable learner for GMMs, settling the agnos-

tic private learnability of GMMs. Finally, our sample complexity improves their result in terms of dependence on the accuracy parameter by a factor of $\frac{1}{\alpha^2}$.

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

## Appendix A. Proof of the main reduction

In this section, we prove our main reduction in Theorem 4.

**Proof** We will first prove the utility and then the privacy.

**Utility analysis.** Let $g$ be the true distribution, $\rho = 0.91$, $\alpha' = \frac{\alpha}{3+4C}$, $\varepsilon' = \frac{\varepsilon}{1+\log(1/\alpha')}$, $\delta' = \frac{\delta}{1+\log(1/\alpha')}$, and $\beta' \in (0,1)$ be a parameter to be set later. We want to $(\varepsilon, \delta)$-privately output a distribution $\hat{f} \in \mathcal{F}$, such that with probability at least $1 - \beta$, $d_{\mathrm{TV}}(\hat{f}, g) \leq 7C \cdot d_{\mathrm{TV}}(g, \mathcal{F}) + \alpha$. We do this in three steps:

**Step I.** Non-private candidate generation:

Let $m_1 \geq m + \frac{\log(L/\beta')}{\alpha'^2}$, and $T \geq \max\{\frac{640}{\varepsilon'}\log(\frac{1280\,L}{\beta'\delta'\varepsilon'^2}), \frac{20}{\varepsilon'}\log(1 + \frac{e^{\varepsilon'}}{2\delta'})\}$. Consider $T$ disjoint data sets $D_1, D_2, \cdots, D_T$ each of size $m_1$ drawn i.i.d. from $g$. For each data set $D_i$, use $m$ samples to run the $(C, \alpha')$-accurate $(m, \rho, L)$-list-globally-stable learner for $\mathcal{F}$. Let $\mathcal{L}_i$ be the outputted list. From Definition 3, we know that there exists a distribution $\tilde{g}$ satisfying $\mathbb{P}_{S \sim g^m}[\tilde{g} \in \mathcal{L}_i] \geq \rho$ and $d_{\mathrm{TV}}(\tilde{g}, g) \leq C \cdot d_{\mathrm{TV}}(g, \mathcal{F}) + \alpha'$. Let $X_i = \mathbf{1}\{\tilde{g} \in \mathcal{L}_i\}$ be a binary random variable. Then using Hoeffding's inequality, we have

$$\mathbb{P}\left[\sum_{i \in [T]} X_i \leq \mathbb{E}\left[\sum_{i \in [T]} X_i\right] - t\right] \leq \exp(-\frac{2t^2}{T}).$$

Substituting $t = 0.01T\rho$ results in $\mathbb{P}\left[\sum_{i \in [T]} X_i \leq 0.99T\rho\right] \leq \exp(-2.10^{-4}T\rho^2)$. Since $T \geq \frac{\log(1/\beta')}{\rho^2}$, we get that with probability at least $1 - \beta'$, $\tilde{g}$ is in at least $0.99\rho$ fraction of lists. Let the indices of such lists be the set $\mathcal{I} \subseteq [T]$.

---

**Algorithm 2:** Private Agnostic Density Estimation

---

**Input:** Parameters $\alpha, \beta, \delta \in (0,1), \varepsilon > 0, C \geq 1, m, L \in \mathbb{N}$, a $(C, \frac{\alpha}{3+4C})$-accurate $(m, 0.91, L)$-list-globally-stable learner SLD for $\mathcal{F}$.

**Output:** A distribution $f \in \mathcal{F}$ (or $\perp$)

1 Let $\alpha' = \frac{\alpha}{3+4C}, \varepsilon' = \frac{\varepsilon}{1+\log(1/\alpha')}, \delta' = \frac{\delta}{1+\log(1/\alpha')}, \beta' = \frac{\beta\varepsilon'}{7680 \log(\frac{9830400L}{\varepsilon'^3 \beta \delta'}) \log(1/\alpha')}$,

$m_1 = m + \frac{\log(L/\beta')}{\alpha'^2}$, and $T = \frac{640}{\varepsilon'} \log(\frac{1280\,L}{\beta'\delta'\varepsilon'^2})$.

2 Draw a sample set $D$ of size $T \cdot m_1$, and randomly partition it to $T$ disjoint data sets $D_1, D_2, \cdots, D_T$.

**for** $i \in [T]$ **do**

3 $\quad$ Randomly partition $D_i$ into two disjoint subsets $D_i^1$ of size $m$, and $D_i^2$ of size $\frac{\log(L/\beta')}{\alpha'^2}$.

4 $\quad$ $\mathcal{L}_i = \text{SLD}(D_i^1)$ // Run the list globally stable learner on $D_i^1$.

5 $\quad$ $\hat{f}_i = \text{MDE}(\mathcal{L}_i, D_i^2, \alpha', \beta')$ // Find a ``good'' distribution in $\mathcal{L}_i$ using MDE algorithm on $D_i^2$.

**end**

/* Filter $\mathcal{L}_i$'s to get ``good'' candidates along with their ``stability'' as their scores. */

6 $(\mathcal{M}, \text{score}) = \text{Get-Candidates-And-Scores}(\alpha', \delta', \varepsilon', C, T, D, \{\hat{f}_1, \cdots, \hat{f}_T\}, \{\mathcal{L}_1, \cdots, \mathcal{L}_T\})$

/* Run Choosing Mechanism to select a ``good'' and ``stable'' candidate. */

7 **return** Choosing-Mechanism$(\mathcal{M}, D, \text{score}, \beta', \varepsilon', \delta', L)$

---

---

**Algorithm 3:** Get Candidates And Scores

---

**Input:** Parameters $\alpha', \delta' \in (0,1), \varepsilon' > 0, C \geq 1, T \in \mathbb{N}$, a data set $D \in \mathcal{X}^*$, a set of distributions $\{\hat{f}_1, \hat{f}_2, \cdots, \hat{f}_T\} \in \mathcal{F}^T$, and a set of lists $\{\tilde{\mathcal{L}}_1, \tilde{\mathcal{L}}_2, \cdots, \tilde{\mathcal{L}}_T\} \in (\mathcal{F}^*)^T$.

**Output:** A set of filtered distributions $\mathcal{M} \subseteq \mathcal{F}$, and a score function $\text{score} : \mathcal{M} \to \mathbb{R}$.

1 Let $LP = 0, UP = 1$.

2 Let $\text{state} = \emptyset, \text{score} = \emptyset, \mathcal{M} = \emptyset$.

**while** $UP - LP > \alpha'$ **do**

3 $\quad$ Let $\widetilde{\text{OPT}} = \frac{LP+UP}{2}$

4 $\quad$ Let $(\text{state}, \mathcal{M}, \text{score}) =$

$\quad$ Filter-And-Test$(\alpha', \delta', \widetilde{\text{OPT}}, \varepsilon', C, T, D, \{\hat{f}_1, \cdots, \hat{f}_T\}, \{\mathcal{L}_1, \cdots, \mathcal{L}_T\})$

$\quad$ **if** *state = "reject"* **then**

5 $\quad\quad$ Let $LP = \widetilde{\text{OPT}}$

$\quad$ **else**

6 $\quad\quad$ Let $UP = \widetilde{\text{OPT}}$

$\quad$ **end**

**end**

7 **return** $(\mathcal{M}, \text{score})$

---

---

**Algorithm 4:** Filter and Test

---

**Input:** Parameters $\alpha', \delta', \widetilde{\text{OPT}} \in (0,1)$, $\varepsilon' > 0$, $C \geq 1$, $T \in \mathbb{N}$, a data set $D \in \mathcal{X}^*$, a set of distributions $\{\hat{f}_1, \hat{f}_2, \cdots, \hat{f}_T\} \in \mathcal{F}^T$, and a set of lists $\{\mathcal{L}_1, \mathcal{L}_2, \cdots, \mathcal{L}_T\} \in (\mathcal{F}^*)^T$.

**Output:** State $\in \{$"reject", "accept"$\}$, a set of filtered distributions $\mathcal{M} \subseteq \mathcal{F}$, and a score function score $: \mathcal{M} \to \mathbb{R}$.

  **for** $i \in [T]$ **do**

1      $\tilde{\mathcal{L}}_i = \{f \in \mathcal{L}_i : \mathrm{d}_{\mathbf{TV}}(f, \hat{f}_i) \leq 4C \cdot \widetilde{\text{OPT}} + 2\alpha'\}$ // Filter out ``bad'' distributions in $\mathcal{L}_i$

  **end**

2 Let $\mathcal{M} = \bigcup_{i \in [T]} \tilde{\mathcal{L}}_i$

  **for** $f \in \mathcal{M}$ **do**

    /* The score of each element $f \in \mathcal{M}$ is defined w.r.t. the data set $D$ which is implicit in the construction of $\tilde{\mathcal{L}}_i$'s */

3      $\mathrm{score}(f, D) = |\{i \in [T] : f \in \tilde{\mathcal{L}}_i\}|$.

  **end**

4 Let $\mathrm{MAX} = \max_{f \in \mathcal{M}} \mathrm{score}(f, D)$.

5 Let $\widetilde{\mathrm{MAX}} = \mathrm{MAX} + \mathrm{TLap}(1, \varepsilon', \delta')$.

  **if** $\widetilde{MAX} < 0.8T + \frac{1}{\varepsilon'} \log(1 + \frac{e^{\varepsilon'}}{2\delta'})$ **then**

6     **return** ("reject", $\mathcal{M}$, score $: \mathcal{M} \to \mathbb{N}$)

  **else**

7     **return** ("accept", $\mathcal{M}$, score $: \mathcal{M} \to \mathbb{N}$)

  **end**

---

**Step II.** Non-private candidate filtering:

Recall that, the size of each data set $D_i$ is at least $m_1 \geq m + \frac{\log(L/\beta')}{\alpha'^2}$. Now, we use the remaining $\frac{\log(L/\beta')}{\alpha'^2}$ samples from $D_i$ to filter out "bad" distributions in $\mathcal{L}_i$ w.r.t. $d_{TV}$. The filtering process works as follows; Theorem 9 (MDE) implies that for every $i \in \mathcal{I}$, using $\frac{\log(L/\beta')}{\alpha'^2}$ samples, we are able to find a distribution $\hat{f}_i \in \mathcal{L}_i$ such that w.p. at least $1 - \beta'$,

$$d_{TV}(\hat{f}_i, g) \leq 3\, d_{TV}(g, \mathcal{L}_i) + \alpha' \tag{1}$$
$$\leq 3(d_{TV}(g, \tilde{g}) + d_{TV}(\tilde{g}, \mathcal{L}_i)) + \alpha' \tag{2}$$
$$\leq 3(C \cdot d_{TV}(g, \mathcal{F}) + 0) + \alpha' \tag{3}$$
$$\leq 3C \cdot d_{TV}(g, \mathcal{F}) + \alpha' \tag{4}$$

Now we remove any distribution $f$ from $\mathcal{L}_i$ that is far from $\hat{f}_i$. The filtering radius depends on a value which we call $\widetilde{\text{OPT}}$. In Algorithm 3, we show how to choose a suitable filtering radius. We will later describe this process. For now, let the filtered list to be:

$$\tilde{\mathcal{L}}_i = \{f \in \mathcal{L}_i : d_{TV}(f, \hat{f}_i) \leq 4C \cdot \widetilde{\text{OPT}} + 2\alpha'\} \tag{5}$$

As a result, we get $T$ lists $\tilde{\mathcal{L}}_1, \cdots, \tilde{\mathcal{L}}_T$ each of size at most $L$. Let $\text{OPT} = d_{TV}(g, \mathcal{F})$. We are not aware of this value since $g$ is unknown. We claim that, if $\text{OPT} \leq \widetilde{\text{OPT}}$, then for every $i \in \mathcal{I}$, $\tilde{\mathcal{L}}_i$ still contains $\tilde{g}$. This is because:

$$d_{TV}(\tilde{g}, \hat{f}_i) \leq d_{TV}(\tilde{g}, g) + d_{TV}(g, \hat{f}_i) \tag{6}$$
$$\leq (C \cdot d_{TV}(g, \mathcal{F}) + \alpha') + (3C \cdot d_{TV}(g, \mathcal{F}) + \alpha') \tag{7}$$
$$\leq 4C \cdot d_{TV}(g, \mathcal{F}) + 2\alpha' \tag{8}$$
$$\leq 4C \cdot \widetilde{\text{OPT}} + 2\alpha' \tag{9}$$

hence $\tilde{g}$ is not filtered from $\mathcal{L}_i$, and we have $\tilde{g} \in \tilde{\mathcal{L}}_i$.

Furthermore, if $\widetilde{\text{OPT}} \leq \text{OPT} + \alpha'$, then for every $i \in \mathcal{I}$, $\tilde{\mathcal{L}}_i$'s members are "good" w.r.t. $d_{TV}$, since for any $f \in \tilde{\mathcal{L}}_i$ it holds that:

$$d_{TV}(f, g) \leq d_{TV}(f, \hat{f}_i) + d_{TV}(\hat{f}_i, g) \tag{10}$$
$$\leq (4C \cdot \widetilde{\text{OPT}} + 2\alpha') + (3C \cdot d_{TV}(g, \mathcal{F}) + \alpha') \tag{11}$$
$$\leq 7C \cdot \text{OPT} + (3 + 4C)\alpha'. \tag{12}$$

Let the "score" function be defined as $\text{score}(f, D) = |\{i \in [T] : f \in \tilde{\mathcal{L}}_i\}|$ (see Algorithm 4). We have:

- $\text{score}(f, \emptyset) = 0$ for all $f \in \mathcal{F}$.

- If $D' = D \cup \{\mathcal{L}\}$, then $\text{score}(f, D) + 1 \geq \text{score}(f, D') \geq \text{score}(f, D)$ for all $f \in \mathcal{F}$. Since each list $\mathcal{L}_i \in D$ contributes to any $f$'s score by at most 1.

- There are at most $k = L$ many distributions $f \in \mathcal{F}$ such that $\text{score}(f, D) + 1 = \text{score}(f, D')$.

At a high level, the score of each distribution (candidate) represents its "stability" with respect to the filtered lists. We will use this score function to privately select a distribution (candidate) with high score using the Choosing Mechanism 1.

From Eq. 9, we can say that if $\text{OPT} \leq \widetilde{\text{OPT}}$, then $\tilde{g}$ is still in a large fraction of the lists and receives a large score. At this point, we privately check whether the maximum score is large enough; this will be used to determine if we have chosen a suitable filtering radius $\widetilde{\text{OPT}}$. If the filtering radius is small, then the maximum score would be low (and we cannot claim the output is indeed a "good" distribution. Recall that for $i \in [T] \setminus \mathcal{I}$, members of $\tilde{\mathcal{L}}_i$ may be "bad" and have low scores.). On the other hand, if we choose the filtering radius to be large, the maximum score would be large, but at the same time, it will affect the utility of the algorithm since we are not completely filtering "bad" distributions from the $\mathcal{L}_i$'s where $i \in \mathcal{I}$ (which potentially can have large scores). In Algorithm 3, we show how to adaptively choose a suitable $\widetilde{\text{OPT}}$ using the Filter-And-Test subroutine 4.

From Eq. 9, recall that, if $\text{OPT} \leq \widetilde{\text{OPT}}$, then for every $i \in \mathcal{I}$, $\tilde{\mathcal{L}}_i$ contains $\tilde{g}$, and we have $\text{MAX} \geq \text{score}(\tilde{g}, D) \geq 0.99\rho T > 0.9T$ (with probability at least $1 - 2T\beta'$ over the correctness of the stable list decoding algorithm and MDE). Using the fact that $T \geq \frac{20}{\varepsilon'} \log(1 + \frac{e^{\varepsilon'}}{2\delta'})$, we get that with probability 1, $\widetilde{\text{MAX}} \geq \text{MAX} - |\text{TLap}(1, \varepsilon', \delta')| > 0.8T + |\text{TLap}(1, \varepsilon', \delta')| = 0.8T + \frac{1}{\varepsilon'} \log(1 + \frac{e^{\varepsilon'}}{2\delta'})$ (see Definition 14). This means that for all values of $\widetilde{\text{OPT}}$ that satisfy $\text{OPT} \leq \widetilde{\text{OPT}}$, the Filter-And-Test subroutine 4 does not output "reject" (with probability at least $1 - 2T\beta'$).

Now using a binary search technique in Algorithm 3, we are able to find a filtering value $\widetilde{\text{OPT}}$, such that the Filter-And-Test subroutine 4 does not output "reject". Putting together with a union bound over $\log(1/\alpha')$ iterations of the binary search algorithm, we can say with probability at least $1 - 2T \log(1/\alpha')\beta'$ we have $LP \leq \text{OPT}$, and $\widetilde{\text{OPT}} \leq \alpha' + \text{OPT}$.

Note that, the above argument does not necessarily imply that $\text{OPT} \leq \widetilde{\text{OPT}}$. However, in the Filter-And-Test subroutine 4, if $\text{MAX} < 0.8T$, then with probability 1, we have $\widetilde{\text{MAX}} < 0.8T + \frac{1}{\varepsilon'} \log(1 + \frac{e^{\varepsilon'}}{2\delta'})$ and the subroutine outputs "reject". Thus, when the Filter-And-Test subroutine 4 does not output "reject", we have $\text{MAX} \geq 0.8T$ which is enough for us to provide utility guarantee. In other words, we still have a reasonably "stable" candidate in the filtered lists (which might be different from $\tilde{g}$).

**Step III.** Private selection:

Now, let $\hat{f} = \text{ChoosingMechanism}(\mathcal{F}, D, \text{score}, \beta', \varepsilon', \delta', k)$[5]. Using Theorem 13 together with a union bound, implies with probability at least $1 - (2T \log(1/\alpha') + 1)\beta'$:

$$\text{score}(\hat{f}, D) \geq \max_{f \in \mathcal{F}} \text{score}(f, D) - \frac{16}{\varepsilon'} \log(\frac{4kT}{\beta'\varepsilon'\delta'}) \geq 0.8T - \frac{16}{\varepsilon'} \log(\frac{4kT}{\beta'\varepsilon'\delta'}) \geq 0.7T.$$

In the last inequality we used Claim 26 with $c_1 = \frac{160}{\varepsilon'}, c_2 = \frac{160}{\varepsilon'} \log(\frac{4L}{\varepsilon'\beta'\delta'})$ to get $\frac{16}{\varepsilon'} \log(\frac{4kT}{\beta'\varepsilon'\delta'}) \leq 0.1T$.

Thus, with probability at least $1 - (2T \log(1/\alpha') + 1)\beta' \geq 1 - 3T \log(1/\alpha')\beta'$, we have

$$\text{score}(\hat{f}, D) = |\{i \in [T] : \hat{f} \in \tilde{\mathcal{L}}_i\}| \geq 0.7T$$

Now that we chose a suitable filtering radius satisfying $\widetilde{\text{OPT}} \leq \alpha + \text{OPT}$, using Eq. 12, we know that for $i \in \mathcal{I}$, $\tilde{\mathcal{L}}_i$'s members are "good" w.r.t $d_{\text{TV}}$. This means that, at most $\frac{|[T] \setminus \mathcal{I}|}{T} \leq 0.01\rho \leq 0.001$

---

5. In the last line of the Algorithm 2, we used $\mathcal{M} \subseteq \mathcal{F}$ as the set of candidates, which is algorithmically more efficient. The reason is that the Choosing Mechanism only considers candidates with non-zero scores, which are the members of $\mathcal{M}$. However, in the analysis, for compatibility with Theorem 13, we consider the set of all candidates $\mathcal{F}$.

fraction of lists may contain "bad" distributions w.r.t. $\mathrm{d_{TV}}$. This implies that any "bad" distribution could have the score of at most $0.001T$. Therefore, $\hat{f}$ whose score is at least $0.7T$, is indeed a "good" distribution and for some $i \in \mathcal{I}$, $\hat{f}$ belongs to $\tilde{\mathcal{L}}_i$:

$$\mathrm{d_{TV}}(\hat{f}, g) \leq \max_{f \in \tilde{\mathcal{L}}_i} \mathrm{d_{TV}}(f, g) \leq 7C \cdot \mathrm{OPT} + (3 + 4C)\alpha'. \tag{13}$$

**Privacy analysis.** Note that:

- All iterations of the binary search in Algorithm 3 are $(\varepsilon', \delta')$-DP because of the privacy guarantee of Truncated Laplace Mechanism (see Theorem 15), and the fact that the sensitivity of the score function is 1.

- The score function satisfies all properties in Theorem 13, which implies that the Choosing Mechanism used in Algorithm 2 is also $(\varepsilon', \delta')$-DP.

Putting together, the Algorithm 2 is $((1 + \log(1/\alpha))\varepsilon', (1 + \log(1/\alpha))\delta')$-private due to the composition property of differential privacy (see Lemma 12).

Now, we substitute $\alpha' = \frac{\alpha}{3+4C}, \beta' = \frac{\beta\varepsilon'}{7680 \log(\frac{9830400L}{\varepsilon'^3 \beta \delta'}) \log(1/\alpha')}, \varepsilon' = \frac{\varepsilon}{1+\log(1/\alpha')}$, and $\delta' = \frac{\delta}{1+\log(1/\alpha')}$. Final calculations (see Claim 26[6]), implies that with probability at least $1 - 3T\log(1/\alpha)\beta' \geq 1 - \beta$, it holds that $\mathrm{d_{TV}}(\hat{f}, g) \leq 7C \cdot \mathrm{d_{TV}}(g, \mathcal{F}) + \alpha$. The total sample complexity is:

$$T \cdot m_1 = \frac{640}{\varepsilon'} \log(\frac{1280\,L}{\beta'\delta'\varepsilon'^2}) \cdot (m + \frac{\log(L/\beta')}{\alpha'^2}) = \tilde{O}\left(\frac{\log(L/\delta\beta)}{\varepsilon} \cdot (m + \frac{\log(L/\beta)}{\alpha^2})\right).$$

■

## Appendix B. Missing proofs from Section 4

The following lemma is useful in the construction of our list globally stable learner for mixtures. At a high level, it states that if a distribution $g$ is close to a class of mixtures $k$-mix$(\mathcal{F})$, then $g$ can be expressed as a mixture, such that the overall mass of far components from class $\mathcal{F}$ is small.

**Lemma 23 (Lemma 7 of Ashtiani et al. (2018b))** *Let $\mathcal{F}$ be a class of distributions, then any distribution $g$ can be written as a mixture $g = \sum_{i\in[k]} w_i g_i$ such that $w_i \geq 0$, $\sum_{i\in[k]} w_i = 1$, and $g_i$'s are distributions satisfying $\sum_{i\in[k]} w_i \, \mathrm{d_{TV}}(g_i, \mathcal{F}) = \mathrm{d_{TV}}(g, k\text{-mix}(\mathcal{F}))$.*

**Proof of Theorem 16:**
**Proof** Let $g$ be the true distribution. Let $g = \sum_{i\in[k]} w_i g_i$ be the representation from Lemma 23 satisfying $\sum_{i\in[k]} w_i \, \mathrm{d_{TV}}(g_i, \mathcal{F}) = \mathrm{d_{TV}}(g, k\text{-mix}(\mathcal{F}))$. Let $\mathcal{A}_1$ be the $(C, \alpha)$-accurate $(m, 1-\beta, L)$-list-globally-stable learner for $\mathcal{F}$. Let $S$ be an i.i.d. sample set from $g$ with size $m_1$. Define $\mathcal{L} = \{\sum_{j\in[k]} w_j h_j : w \in \hat{\Delta}_k, h_j \in \mathcal{H}\}$, where $\mathcal{H} = \bigcup_{\tilde{S}\subseteq S:|\tilde{S}|=m} \mathcal{A}_1(\tilde{S})$, and $\hat{\Delta}_k$ is an $\frac{\alpha}{k}$-cover for the $(k-1)$-dimensional probability simplex w.r.t. $\ell_\infty$ from Claim 24. Consider an algorithm $\mathcal{A}_2$ that receives the i.i.d. sample set $S$, and outputs $\mathcal{L}$. We claim that $\mathcal{A}_2$ is a $(C, 5\alpha)$-accurate $(m_1, 1 - 2k\beta, L_1)$-list-globally-stable learner for $k$-mix$(\mathcal{F})$.

---

6. With $c_1 = \frac{1920}{\varepsilon'\beta}, c_2 = \frac{c_1}{\beta} \log(\frac{1280L}{\varepsilon'^2\delta'})$.

Define $I = \{i \in [k] : w_i \geq \frac{\alpha}{k}\}$. For every $i \in I$, we can write $g = w_i g_i + (1 - w_i) \sum_{j \neq i} \frac{w_j}{1 - w_i} g_j$. After receiving $N \geq m_1$ samples from $g$, the number of samples coming from $g_i$ has a binomial distribution. Let the corresponding random variable be $X_N$. Since $w_i \geq \frac{\alpha}{k}$, we have $\mathbb{E}[X_N]/2 \geq m$ and $\mathbb{E}[X_N] \geq 8 \log(1/\beta)$. Using the Chernoff bound (Theorem 4.5(2) of (Mitzenmacher and Upfal, 2005)), we have $\mathbb{P}[X_N \leq m] \leq \mathbb{P}[X_N \leq \mathbb{E}[X_N]/2] \leq \exp(-\mathbb{E}[X_N]/8) \leq \beta$. Meaning that after drawing $N \geq m_1$ samples from $g$, with probability at least $1 - \beta$, we will have $m$ samples coming from $g_i$. Using a union bound, with probability at least $1 - k\beta$, for all $i \in I$, there exists $m$ samples coming from $g_i$. Thus, using the fact that $\mathcal{A}_1$ is a $(m, 1 - \beta, L)$-list-globally-stable learner for $\mathcal{F}$, we can with probability at least $1 - 2k\beta$, for all $i \in I$, there exists a distribution $\tilde{g}_i \in \mathcal{H}$ satisfying $\mathrm{d}_{\mathrm{TV}}(\tilde{g}_i, g_i) \leq C \cdot \mathrm{d}_{\mathrm{TV}}(g_i, \mathcal{F}) + \alpha$. For $i \in I$, let $\tilde{w}_i = \frac{1}{\sum_{j \in I} w_j} w_i$. Note that there exists a $w^* \in \hat{\Delta}_k$, such that $|w_i^* - \tilde{w}_i| \leq \alpha/k$ (for $i \in [k] \setminus I$, we set $\tilde{w}_i = w_i^* = 0$). Define $\tilde{g} = \sum_{i \in [k]} w_i^* \tilde{g}_i$ (for $i \in [k] \setminus I$, we set $\tilde{g}_i$ to be an arbitrary element in $\mathcal{H}$).

Now, we are ready to verify the two properties of list globally stable learning (recall Definition 3);

**Property I**. $\mathcal{A}_2$ is a list globally stable algorithm.

By construction, $\tilde{g} \in \mathcal{L}$ with probability at least $1 - 2k\beta$. Also, note that

$$|\mathcal{L}| = |\hat{\Delta}_k||\mathcal{H}|^k = (\frac{k}{\alpha})^k \left( L \binom{m_1}{m} \right)^k \leq (\frac{k}{\alpha})^k L^k \left( \frac{2mke + 8ke \log(1/\beta)}{\alpha m} \right)^{mk} \tag{14}$$

$$\leq (\frac{Lk}{\alpha})^k \left( \frac{10ek \log(1/\beta)}{\alpha} \right)^{mk}. \tag{15}$$

**Property II**. $\mathcal{A}_2$ preserves agnostic utility guarantee:

$$\mathrm{d}_{\mathrm{TV}}(\tilde{g}, g) \leq \mathrm{d}_{\mathrm{TV}}(\sum_{i \in [k]} w_i^* \tilde{g}_i, \sum_{i \in [k]} w_i g_i) \leq \frac{1}{2} \sum_{i \in [k]} ||w_i^* \tilde{g}_i - w_i g_i||_1 \tag{16}$$

$$\leq \frac{1}{2} \sum_{i \in [k]} ||w_i^* \tilde{g}_i - w_i \tilde{g}_i||_1 + \frac{1}{2} \sum_{i \in [k]} ||w_i \tilde{g}_i - w_i g_i||_1 \tag{17}$$

$$\leq \sum_{i \in [k]} |w_i^* - w_i| + \sum_{i \in [k]} w_i \, \mathrm{d}_{\mathrm{TV}}(\tilde{g}_i, g_i) \tag{18}$$

$$\leq \sum_{i \in [k]} |w_i^* - w_i| + \sum_{i \in I} w_i \, \mathrm{d}_{\mathrm{TV}}(\tilde{g}_i, g_i) + \sum_{i \in [k] \setminus I} w_i \, \mathrm{d}_{\mathrm{TV}}(\tilde{g}_i, g_i) \tag{19}$$

$$\leq \sum_{i \in [k]} |w_i^* - w_i| + \sum_{i \in I} w_i \left( C \cdot \mathrm{d}_{\mathrm{TV}}(g_i, \mathcal{F}) + \alpha \right) + \sum_{i \in [k] \setminus I} \frac{\alpha}{k} \, \mathrm{d}_{\mathrm{TV}}(\tilde{g}_i, g_i). \tag{20}$$

Recall that, $\sum_{i \in I} w_i \, d_{TV}(g_i, \mathcal{F}) \leq \sum_{i \in [k]} w_i \, d_{TV}(g_i, \mathcal{F}) = d_{TV}(g, k\text{-mix}(\mathcal{F}))$. Therefore, we can write:

$$d_{TV}(\tilde{g}, g) \leq \sum_{i \in [k]} |w_i^* - w_i| + C \cdot d_{TV}(g, k\text{-mix}(\mathcal{F})) + 2\alpha \tag{21}$$

$$\leq \sum_{i \in [k]} |w_i^* - \tilde{w}_i| + \sum_{i \in [k]} |\tilde{w}_i - w_i| + C \cdot d_{TV}(g, k\text{-mix}(\mathcal{F})) + 2\alpha \tag{22}$$

$$\leq \sum_{i \in [k]} \frac{\alpha}{k} + \sum_{i \in [k]} |\tilde{w}_i - w_i| + C \cdot d_{TV}(g, k\text{-mix}(\mathcal{F})) + 2\alpha \tag{23}$$

$$\leq C \cdot d_{TV}(g, k\text{-mix}(\mathcal{F})) + 3\alpha + \sum_{i \in I} w_i \left( \frac{1}{\sum_{j \in I} w_j} - 1 \right) + \sum_{i \in [k] \setminus I} w_i \tag{24}$$

$$= C \cdot d_{TV}(g, k\text{-mix}(\mathcal{F})) + 3\alpha + \left( 1 - \sum_{i \in I} w_i \right) + \sum_{i \in [k] \setminus I} w_i \tag{25}$$

$$= C \cdot d_{TV}(g, k\text{-mix}(\mathcal{F})) + 3\alpha + 2 \sum_{i \in [k] \setminus I} \frac{\alpha}{k} \tag{26}$$

$$\leq C \cdot d_{TV}(g, k\text{-mix}(\mathcal{F})) + 5\alpha. \tag{27}$$

■

## Appendix C. Missing proofs from Section 5

Here, we formally prove Theorem 22.

**Proof** Corollary 21 implies that $k\text{-mix}(\mathcal{G}_d)$ is $(3, 5\alpha)$-accurate $(m_1, 0.91, L_1)$-list-globally-stable learnable, where $L_1 = (\frac{Lk}{\alpha})^k \left( \frac{10ek \log(k/0.045)}{\alpha} \right)^{mk}$, $m_1 = \frac{2mk + 8k \log(k/0.045)}{\alpha}$, $L = (d \log(k/0.045))^{O(d^2 \log(d/\alpha))}$, and $m = O(d \log(k/0.045))$ . Putting together with Theorem 4, we get that $k\text{-mix}(\mathcal{G})$ is $(\varepsilon, \delta)$-DP 21-agnostic $(n, \alpha, \beta)$-learnable with

$$n = \tilde{O} \left( \frac{kd^2 + \log(1/\beta\delta)}{\varepsilon} \cdot \frac{kd^2 + \log(1/\beta)}{\alpha^2} \right) \tag{28}$$

$$= \tilde{O} \left( \frac{k^2 d^4 + kd^2 \log(1/\delta\beta) + \log^2(1/\beta)}{\alpha^2 \varepsilon} \right) \tag{29}$$

samples.

■

## Appendix D. Additional facts

The following simple proposition gives a finite cover for weight vectors used to construct a mixture.

**Claim 24** *Let $\alpha \in (0, 1]$. There is an $\alpha$-cover for the $(k-1)$-dimensional probability simplex $\{(w_1, w_2, ..., w_k) \in \mathbb{R}_{\geq 0}^k : \sum_{i \in [k]} w_i = 1\}$ w.r.t. $\ell_\infty$ of size at most $(1/\alpha)^k$.*

**Proof** Partition the cube $[0,1]^k$ into small cubes of side-length $1/\alpha$. If for a cube $c$, we have $c \cap \Delta_k \neq \emptyset$, put one arbitrary point from $c \cap \Delta_k$ into the cover. The size of the constructed cover is no more than $(1/\alpha)^k$ which is the total number of small cubes. ∎

**Claim 25** *Let $x \geq 1$. Then $1 + \frac{\log 2}{x} + \frac{\log x}{x} < 2$.*

**Proof** Let $f(x) = 1 + \frac{\log 2}{x} + \frac{\log x}{x}$. Then $f'(x) = -\frac{\log 2}{x^2} + \frac{1 - \log x}{x^2} = \frac{1 - \log(2x)}{x^2}$. Note that $f'(x)$ is decreasing so $f$ is concave. In addition, $x = e/2$ is the only root of $f'$ so $f$ is maximized at $e/2$. Thus, $f(x) \leq f(e/2) = 1 + \frac{2}{e} < 2$. ∎

**Claim 26** *Let $c_1 \geq e/2, c_2 > 0$. If $x \geq 4c_1 \log(2c_1) + 2c_2$, then $x \geq c_1 \log(x) + c_2$.*

**Proof** If $x \geq 4c_1 \log(2c_1) + 2c_2$, then $\frac{x}{2} \geq c_2$. It is sufficient to show $x \geq 2c_1 \log(x)$. Consider the function $f(x) = x - 2c_1 \log(x)$. Then $f'(x) = 1 - \frac{2c_1}{x}$, which implies that for $x > 2c_1$ the function is increasing. As a result, for $x \geq 4c_1 \log(2c_1)$ we have $f(4c_1 \log(2c_1)) = 4c_1 \log(2c_1) - 2c_1 \log(4c_1 \log(2c_1)) = 4c_1 \log(2c_1) - 2c_1 \log(2c_1)[1 + \frac{\log(2)}{\log(2c_1)} + \frac{\log(\log(2c_1))}{\log(2c_1)}] > 0$. The last inequality follows from Claim 25 with $x = \log(2c_1) \geq 1$. Putting together, results in $x \geq c_1 \log(x) + c_2$. ∎

## Appendix E. Discussion on stability

Various notions of stability have been proposed in the context of differential privacy. Global stability (Thakurta and Smith, 2013; Bun et al., 2020) is one such notion.

**Definition 27 (Global stability (Bun et al., 2020))** *Let $m \in \mathbb{N}, \rho > 0$. We say an algorithm $\mathcal{A}$ is $(m, \rho)$-globally-stable if for every distribution $\mathcal{D}$ over input, there exists a hypothesis $h_\mathcal{D}$ such that $\mathbb{P}_{S \sim \mathcal{D}^m}[\mathcal{A}(S) = h_\mathcal{D}] \geq \rho$.*

Global stability requires the algorithm to output *the exact same hypothesis* (with probability $\rho$) when run on i.i.d. data sets. Note that an algorithm that ignores the data set and always outputs the same hypothesis is trivially globally stable. However, we are looking for an algorithm that is both globally stable and accurate (i.e., $h_\mathcal{D}$ should be a "good" hypothesis).

Globally stable algorithms are easy to privatize (e.g., by running them on $O(1/\rho)$ separate i.i.d. sets and using a private histogram (Bun et al., 2019b) to aggregate the results). For binary classification, it has been shown that this stringent definition of stability is achievable for any online learnable class, establishing the equivalence between online learnability and private learnability (Alon et al., 2022). However, this definition is not suitable for for high-dimensional estimation tasks. For example, consider the simple task of mean estimation for a $d$-dimensional Gaussian distribution. Even after discretizing the space of solutions, $\rho$ will be exponentially small in $d$ for any mean estimator that uses $\text{poly}(d)$ samples.

Observing that $\rho$ cannot be generally boosted for globally stable algorithms, Chase et al. (2023) defined the notion of *list replicability*. Instead of requiring outputting the exact same hypothesis, a list replicabile algorithm can output any member from a fixed (but distribution-dependent) list of outcomes.

**Definition 28 (List replicability (Chase et al., 2023))** *Let $m, L \in \mathbb{N}, \rho > 0$. An algorithm $\mathcal{A}$ is called $(m, \rho, L)$-list-replicable if for every distribution $\mathcal{D}$ over input, there exists a set of hypotheses $H_{\mathcal{D}} = \{h_1, h_2, \cdots, h_L\}$ such that $\mathbb{P}_{S \sim \mathcal{D}^m} [\mathcal{A}(S) \in H_{\mathcal{D}}] \geq \rho$.*

For the simple task of mean estimation, it is possible design a list replicable algorithm that uses a small number of samples ($m =$poly$(d)$) and with high probability of success ($\rho$ close to 1) albeit with an exponential dependence of $L$ on the dimension. While list replicability has been found useful for studying the relationship between privacy and algorithmic replicability (Impagliazzo et al., 2022; Chase et al., 2023), it does not seem to be suitable for designing private density estimators: even if we have a list replicable algorithm for learning a class like GMMs, turning it into a private algorithm can blow up the sample complexity. To see this, recall that $L$ could be very large (e.g., exponentially large in $d$). Therefore, one would have to run the non-private algorithm on many (distinct) data sets to start seeing repetitive outcomes (i.e., "collisions"). However, the sample complexity of such an approach will be quite poor, and it is not clear how to privatize a list replicable algorithm otherwise.

To overcome this challenge, we utilize the related notion of list global stability, which was implicitly used in Ghazi et al. (2021a) and later formally defined in Ghazi et al. (2021b).

