# OpenReview forum: "Agnostic Private Density Estimation for GMMs via List Global Stability"
_algorithmiclearningtheory.org/ALT/2025/Conference — ALT 2025_

### Official Review · Reviewer_okap · 2024-11-08
**Private agnostic GMM density estimation**

**Rating:** 7
**Confidence:** 4

**Review:**

This paper upper bounds the sample complexity of differentially private density estimation of Gaussian mixture models (GMMs). It considers agnostic learning: the input is iid from an arbitrary distribution, but we want to compete with the best mixture of $k$ Gaussians in $d$ dimensions. The mixture weights, means, and (positive definite) covariances may be arbitrary. The main result is an $(\varepsilon,\delta)$-DP learner with sample complexity roughly $k^2 d^4 \log(1/\delta)/\varepsilon$ (ignoring accuracy parameters). This is the first result for this task; applied to the realizable setting, it improves upon Afzali et al. (2024).

The analysis constructs "list globally stable" (LGS) learners, which return a set of distributions containing (usually) a good estimate and also satisfy a form of stability: for any input distribution, there is an output that often appears in the list. The proof has three main parts. First, one can turn a non-private LGS learner into a private algorithm. This seems to be a careful application of existing private selection techniques. Second, we see how to construct an LGS learner for mixture models given access to an LGS learner for the single-component/base distribution. The algorithm is simple (run the base learner on every subset of samples) but the analysis seems subtle. Both of these steps apply to general distributions, not just Gaussians. Finally, we construct an LGS learner for Gaussians, using robust compression schemes and discretization.

This is a nice paper, drawing from distribution learning, robustness, and privacy. I think many people at ALT will find it interesting. The authors did a good job giving an overview of the proof, despite the messiness in the problem parameters. The submission's main weakness is that it's a little hard to identify the new technical ideas.

A few smaller points:
- The introduction jumps quickly between "agnostic" and "robust." Sometimes I couldn't follow the conclusions from related work, such as "Therefore, there is no general recipe..." (p2).
- Related: the paper often talks about agnostic learning as learning under corruption (e.g., p4, "the corruption level"). Unless it's made more explicit, this view might confuse readers.
- Is there a naive upper bound using exponentially many samples? If so, I think that might be useful context. If it's not obvious, then that itself is interesting!
- Section 5.1 contains the sentence "The formal proof is given in Section 5.1."

**Paper Award:**

No

---

> ### Author Response · Authors · 2024-11-22
>
> Thanks for the review and the comments! We will address them in the next version. Regarding your question about whether or not there exists a trivial algorithm with exponential sample complexity: that's a really good question. We are not really sure but if we come up with a simple argument, we will add it to the paper. Thanks for pointing this out!

---

### Official Review · Reviewer_PAcj · 2024-11-11

**Rating:** 7
**Confidence:** 4

**Review:**

Summary

This paper studies the problem of private density estimation for mixtures of Gaussians in the agnostic setting. This problem was previously studied by (Afzali et al., 2024) in the realiazable setting but remained open in the agnostic setting.

The paper provides upper bounds on the sample complexity of privately learning GMM in the agnostic setting. Their techniques are inspired from existing work on private supervised learning where private algorithms are designed using algorithms that satisfy different notions of stability. In particular, the algorithms in this paper build on the notion of list global stability.

The authors prove their main result via two main steps: first, they show a reduction from agnostic private density estimation to list globally learnable learning, and then they propose a list globally stable algorithm for density estimation of GMM.

Strengths

Overall I think the paper provides some nice results and techniques that would be of interest to the ALT community. The paper solves an interesting problem, providing the first upper bounds on the sample complexity of private density estimation for mixtures of Gaussians in the agnostic setting. Moreover, the results in the paper also improve the sample complexity in the realizable setting over the prior work of Afzali et al. (2024).


Weaknesses


I have two minor complaints about the paper:

1. Clarity\writing: while most of the paper is very clear and easy to read, I think section 4 needs a little more work: currently, three algorithms are listed in that section without providing some description or intuition to help the reader.

2. While the sample complexity bounds are interesting, they are still achieved by algorithms with exponential runtime (in d) as the list size of the stable algorithm is exponential in d. It would be interesting to see if there is a poly-time algorithm that can solve this problem, potentially with a slightly worse sample complexity.

**Paper Award:**

No

---

> ### Author Response · Authors · 2024-11-22
>
> We thank the reviewer for their review and the positive assessment!
> 1. We wanted to clarify that the algorithm is meant for Section 3 (not Section 4) but in any case we will try to improve the exposition around that part.
> 2. As the reviewer mentioned, our algorithm does run in exponential time. Getting a polynomial time is a very interesting question but is already very difficult in the non-private setting. We suspect that one approach to obtain such results would be to try to reduce the private problem to its non-private setting (e.g. [1, 2]) which would be significantly different from the present paper. This approach would be similar to [3] but would still require additional ideas to remove the separability assumption in that paper.
>
> [1] Bakshi, Diakonikolas, Jia, Kane, Kothari, Vempala. Robustly learning mixtures of k arbitrary Gaussians. STOC 2022.
> [2] Liu, Moitra. Robustly Learning General Mixtures of Gaussians. JACM 2023.
> [3] Arbas, Ashtiani, Liaw. Polynomial Time and Private Learning of Unbounded Gaussian Mixture Models. ICML 2023.

---

### Official Review · Reviewer_kKgD · 2024-11-16

**Rating:** 8
**Confidence:** 4

**Review:**

### Summary
This paper gives a private density estimator for mixtures of k d-dimensional Gaussians in the agnostic setting. The estimator is computationally inefficient and its sample complexity is polynomial in d and k. This is the first estimator for the agnostic setting and has slightly improved sample complexity than the existing estimator for the realizable setting.

### Techniques
To achieve this, the authors prove that: a) there exists a list globally stable learner for Gaussians with d samples and list-size d^{d^2}, b) the class of k-mixtures of any list globally
stable learnable class with m samples and list-size L is also list globally stable learnable with mk samples and list-size L^k, and c) any list globally stable learner can be converted into an agnostic private density estimator with a number of samples that grows poly-logarithmically with the list-size L and linearly with the samples m.

List global stability has been introduced in prior work, which also shows how to convert a list globally stable learner into a private learner with a number of samples that is logarithmic in the list-size L, for the task of classification. The reduction follows the same high-level structure in this paper (run the list globally stable learner on subsamples, filter-out bad candidates from the lists, privately detect repeated candidates in the remaining lists) but the individual parts differ significantly and a new analysis is necessary. The analysis combines known techniques such as propose-test-release, the choosing mechanism, robust compression schemes, and ideas connecting the several parts together.

### Score
Overall this paper gives the first estimator for privately learning mixtures of gaussians in the agnostic settings, which is especially challenging. Specifically, it is a task which does not seem particularly amenable to known techniques in private density estimation. The paper proposes techniques that seem more generally applicable, such as the reduction of private density estimation to list globally stable learning, and the recipe to retrieve list globally stable learners for mixtures of classes.

**Paper Award:**

No

---

> ### Author Response · Authors · 2024-11-22
>
> Thank you for your positive assessment!

---

### Meta-Review · Area_Chair_Gs5m · 2024-12-13

**Recommendation:** Accept
**Confidence:** 4

**Metareview:**

This paper tackles the problem of private density estimation for mixtures of k d-dimensional Gaussians, giving the first solution for the agnostic setting (prior work focused on the realizable setting). The focus here is on the sample complexity, which is shown to be polynomial in d and k (this even slighty improves the sample complexity of prior work in the realizable case). To achive this, the authors build on the notion of list global stability, introduced by prior work. More specifically, the authors present a list globally stable learner for Gaussians, extends it to k-mixtures, and then convert it into an agnostic private density estimator. The proposed algorithm is inefficient (runs in exponential time).

The reviewers agree that the paper addresses an important problem, offers strong contributions, and introduces techniques with broader potential. I recommend acceptance.

**Paper Award:**

No